# Health Emergency Disaster Risk Management of Public Transport Systems: A Population-Based Study after the 2017 Subway Fire in Hong Kong, China

**DOI:** 10.3390/ijerph16020228

**Published:** 2019-01-15

**Authors:** Emily Ying Yang Chan, Zhe Huang, Kevin Kei Ching Hung, Gloria Kwong Wai Chan, Holly Ching Yu Lam, Eugene Siu Kai Lo, May Pui Shan Yeung

**Affiliations:** 1Collaborating Centre for Oxford University and CUHK for Disaster and Medical Humanitarian Response (CCOUC), JC (Jockey Club) School of Public Health and Primary Care, The Chinese University of Hong Kong, Hong Kong, China; huangzhe@cuhk.edu.hk (Z.H.); kevin.hung@cuhk.edu.hk (K.K.C.H.); gloria.chan@cuhk.edu.hk (G.K.W.C.); hollylam@cuhk.edu.hk (H.C.Y.L.); Euglsk@cuhk.edu.hk (E.S.K.L.); may.yeung@cuhk.edu.hk (M.P.S.Y.); 2Nuffield Department of Medicine, University of Oxford, Oxford OX3 7BN, UK; 3François-Xavier Bagnoud Center for Health & Human Rights, Harvard University, Boston, MA 02138, USA; 4Accident and Emergency Medicine Academic Unit, The Chinese University of Hong Kong, Hong Kong, China

**Keywords:** public transport, subway, safety, fire, risk perception, emergency response, Health-EDRM

## Abstract

*Background:* Literature on health emergency disaster risk management (Health-EDRM) for urban public transport safety is limited. This study explored: (i) the confidence in public transport safety, (ii) the relationship between socio-demographic characteristics and risk perception of transport safety and (iii) the association between previous first-aid training and response knowledge. *Method:* This is a population-based cross-sectional telephone survey conducted in March 2017, one month after a major subway incident in Hong Kong. Respondents were randomly selected with the Random Digit Dialing method among Cantonese-speaking population ≥15 years. Sociodemographic information, type of transport used and the corresponding worries, response knowledge and previous first-aid training experience (as a proxy for individual skills in Health-EDRM training proxy) were collected. *Results:* Among the 1000 respondents, 87% used public transport daily. The self-reported confidence in subway safety was 85.6% even after a subway fire accident. Female, those with lower income and people unmarried were more likely to express worry about transport safety. About 46.1–63.2% respondents had the correct fire related health response knowledge. Previous first-aid training (32%) was found to be associated with fire response knowledge in a mixed pattern. *Conclusions:* Despite inadequacy in fire response knowledge, previous first-aid training appeared to be a beneficial factor for emergency response knowledge. Emergency responses education should be provided to the public to reduce health losses during emergencies.

## 1. Introduction

Global urbanization has led to the rapid development of public transportations in cities. Subways or metro systems are identified as a recommended mode of urban public transportation as those networks will increase population mobility, geographic connections, and reduce environmental impact from air pollution caused by automobiles [1]. Thus, urban metro systems have an important role in the socio-economic development of many active developing metropolises. About 22% of the world’s 632 largest cities have developed metro systems, and 53 cities in the Asian region has the fasting growing infrastructure with predominant number metro systems when compared with 40 European, 30 North American, 14 South American and one African cities [2].

The heavy reliance of urban residents on metro systems potentially has major implications for health risks and public safety. Globally, numerous critical incidents in urban metro system, such as fire, have been reported to cause massive human impact in high-density cities [1,3,4]. For example, the 1987 King’s Cross Fire in London, the 1990 subway fire in New York City, the 1995 Baku Metro Fire in Azerbaijan, and the 2003 Daegu Subway Fire in South Korea have all resulted in more than 100 casualties [5,6,7]. During emergencies, appropriate personal response may lower vulnerability health risks and even save lives [8]. According to the Sendai Framework, understanding disaster risk and enhancing preparedness are the priorities in risk reduction strategies [9]. Vulnerability is one of the key components in risk assessment [10] while sociodemographic characteristics have been recognized as underlying disaster health risk drivers [10]. Meanwhile, training, which is associated with awareness raising and knowledge enhancement, is defined as non-structural measures in disaster preparedness and resilience [10]. Studies have indicated urban population tends to misjudge their own actual health risk for disaster and emergency [11,12,13,14,15]. Ensuring public safety, education, and emergency preparedness will thus be immensely important to reduce potential harm during and immediately after an emergency incident. For example, an effective railway passenger evacuation during an onboard fire in Shanghai had resulted in no casualties in 2018 [16]. Better understanding of community’s capacity to manage health risks will help to support evidence-based health emergency disaster risk management (Health-EDRM) [17,18] policies and bottom-up resilience capacity building.

Hong Kong, a metropolis in southern China, has a 7.4 million urban based population and has developed a metro system, the Mass Transit Railway (MTR), since 1979. With over five million daily trips made on underground subways and overhead railways, public safety is a priority in such a high density environment [19]. On 10 February 2017, a subway firebomb during the evening rush hour in Hong Kong caused 18 injuries and one death [20]. The accident was regarded as the most serious attack incident in 38 years since the commissioning of the MTR [19]. 

A telephone survey study was conducted after the Hong Kong MTR fire accident in February 2017 to understand the health emergency and disaster risk awareness and preparedness towards transport-related incidents. This study aimed to examine individual’s emergency response and its possible associating factors such as risk perception, previous trainings and other personal characteristics. Factors that could improve personal emergency response and hypothesized that different modes of transport (which is related to risk perception), socio-demographic characteristics, and first-aid training may affect the awareness or knowledge of emergency response were identified. In particular, fire response knowledge in health risks was studied since fire was a common hazard reported in previous transport accidents [1,3,4]. This research paper will report study findings of three main study objectives which include: (i) public transport utilization pattern and confidence associated with public transport safety after a major emergency public incident, (ii) the relationship between socio-demographic characteristics with risk perception and expressed worry with public transport system and (iii) if previous first-aid training, as a proxy for individual skills in Health-EDRM, may be associated with fire injury-related response knowledge to assess potential management capacity to response health risks in a public transportation system. The findings will provide evidence for global metropolis when examining health risk perception for public transportation system and will further support public education and disaster risk reduction policy development to address Health-EDRM in these communities.

## 2. Materials and Methods

### 2.1. Collection

This is a population, cross-sectional telephone-based survey study, which was conducted in 2–12 March 2017, within 1 month after a major subway fire incident in Hong Kong. The study population is stratified according to age, gender and area of residence of the 2016 Hong Kong Census and a representative sample was randomly selected with the Random Digit Dialing method (RDD) through computer generation among the Hong Kong Cantonese-speaking population aged above 15. Each interview lasted between 15 and 25 min. The telephone interviews were conducted by trained interviewers from 6 pm to 10 pm on weekdays and from 10 am to 10 pm on weekends to prevent over representation of the unemployed population. Up to five calls were made to each number before it was considered unanswered. Respondents were chosen based on the “last birthday method” which referred to the household member with the birthday closest to the interview date [11,12,13,21]. 

Self-reported information was collected for socio-demographic background (gender, age, area of residence, marital status and education attainment, Comprehensive Social Security Assistance (CSSA) status), and a total of 12 questions (see Appendix A) were asked to identify respondents’ current pattern of daily transportation), risk perception, worry level of transport safety after a major incidence (expressed worry), knowledge and accuracy of fire emergency response to physical injury, first-aid training, and expressed willingness to learn about community disaster preparedness. Specifically, first-aid training was used as a proxy for Health-EDRM training in the community and CSSA status was used as a proxy to examine socio-economic deprivation and its relationship to the study patterns. Three questions were also asked to explore knowledge and accuracies in health risk and response to fire incidents. Question T1 assessed fire-related first-aid knowledge that is commonly included in first-aid hand book [22] (Should room temperature water or ice water and ice be used to treat the burn? (a) ice water/ice cube; (b) room temperature water). Question T2 assessed the knowledge of the use if a fire blanket which have been promoted by the Hong Kong Fire Services Department (If you are in a fire incident setting and you found someone was on fire, how would you use a fire blanket) (a) put out the fire directly with the fire blanket; (b) cover the victim with the fire blanket and ask them to roll until the fire stops). Question T3 assessed the knowledge of the use of a fire distinguisher in a scenario which was rarely found in official fire response materials nor in first-aid hand books. (If there is no fire blanket at the scene, should fire hoses or extinguishers be used on people) (a) Yes; (b) No).

This study was approved by the Survey and Behavioral Research Ethics Committee of The Chinese University of Hong Kong. Verbal consent was sought from each respondent before the interviews.

### 2.2. Analysis

Descriptive statistical analysis and Pearson’s *χ*^2^ test were conducted on the sociodemographic characteristics of the respondents and the demographic characteristics were further compared with the Hong Kong Population Census data in 2016 [23]. Multiple logistic regression models were constructed to examine associations between variables and the research enquiries. Analyses were conducted using R version 3.1.3 (R Foundation for Statistical Computing, Vienna, Austria). All statistical significance was set at α = 0.05.

## 3. Results

### 3.1. Subjects’ Characteristics

A total valid final study sample of 1000 were collected with a study response rate of 64.8%. Figure 1 detailed the data collection algorithm. The total sample was representative in terms of the distribution of gender, area of residence, and marital status as stated in the 2016 Hong Kong population census data (Table 1).

### 3.2. Daily Transport Utilization and Confidence in Transport Safety

Subway (43.9%) and buses (43.2%), were reported to be the two predominant modes of daily public transport in Hong Kong (Table 2). Analysis by age group showed that 15–24 years group regarded subway (62%) as their most preferred daily mode of transport. For non-motor vehicle based transport (walk/cycle), the elderly (10%) were found to be more likely than the younger groups to walk and cycle. Furthermore, people living in Hong Kong Island and those older than 65 would use other transport modes such as tram and taxi more often than other groups.

A Likert scale ranging from 1–6, 1 for the least safe and 6 for the safest, was used to measure respondents’ rating on the safety level of the transport mode they used daily (defined as “perceived safety” below). Private cars were reported as the safest transport mode (mean = 4.83; standard deviation (SD) = 1.20) despite being ranked as the least utilized mode of transport (4.6%). Meanwhile, buses were regarded as the least safe mode (mean = 4.43; SD = 1.13). Subgroup analysis found no statistically significant differences for gender in perceived safety in the modes of daily transport (Figure 2). 

Despite the survey study was conducted within a month after a major subway incident, 85.6% current subway users were satisfied with their transport routine and regarded their choice as safe (as expressed in “perceived safety”, mean = 4.59; SD = 1.11). When respondents were asked whether they were worry about any disasters/ incidents would happen in the transport mode they used daily (defined as “expressed worry” below), about 35.0% (*n* = 348/981) of respondent expressed concern/worry about transport safety.

### 3.3. Association Among Expressed Worry of Disaster/Incident Risk, Type of Transport Used and Socio-Demographic Factors

Multiple logistic regression was used to examine the association between being worried about transport safety and socio-demographic variables including gender, age, education level, area of residence, CSSA status, marital status, as well as the form of daily transport. Being worried about daily transport use was initially regressed with all mentioned variables. Variables that showed an association with being worried (indicated by *p*-value < 0.1) in stage 1 of the model were included in the second stage (final model). The adjusted odds ratios (OR) and the corresponding 95% confidence interval (CI) are shown in Table 3. No statistically significant association was found between concern/worry about public transport and the mode of transport used. Final model, however, indicated the female gender and people who received CSSA were more likely to express worry about disaster/incident occurrence on their daily transport. Meanwhile, married individuals were less likely to expressed worry when compared with their unmarried counterparts.

### 3.4. Association between Previous First-Aid Training and Fire Injury-Related Response Knowledge, and Willingness to Learn

Around 32.0% of respondents have at some point received first-aid training, the proxy variable which is used to describe individual skills in Health-EDRM. Multivariable logistic regression showed that those with a higher educational level were more likely to have received first-aid training (Table 4). About two-third of respondents (*n* = 671/993), were willing to learn about community disaster preparedness. Specifically, people who had expressed worry about transport safety were also more willing to learn (Table 4).

For the fire response questions, 47.0%, 53.8%, and 61.3% answered T1, T3, and T2 correctly respectively (Table 5). For T1, the first-aid related fire response question (“which ice water should not be used to treat the burn”), people with a higher educational level (OR = 2.55; 95% CI: 1.46–4.46) and had previously received first-aid training (OR = 1.97; 95% CI: 1.48–2.62) were more likely to answer correctly. However, people aged 65 or above (OR = 0.51; 95% CI: 0.30–0.89) were less likely to report a correct answer. Meanwhile, for the use of fire blanket (T2), a technical question that has been promoted by the local Fire Service Department, the female gender (OR = 1.61; 95% CI: 1.23–2.10) and people with a higher education level (OR = 2.16; 95% CI: 1.29–3.6) were more likely to answer correctly. For T3, the use of fire extinguisher, a question that is not related to first-aid nor promoted by the Fire Service Department, people aged 65 years or above (OR = 1.65; 95% CI: 1.01–2.69) and married people (OR = 1.40; 95% CI: 1.03–1.91) had higher rates of correct answers. Of note, the female gender (OR = 0.65; 95% CI: 0.50–0.84) and those who had received first-aid training (OR = 0.73; 95% CI: 0.55–0.96) were less likely to answer T3 correctly. 

In addition, 68.0% of the respondents said they did not know what to do when a fire incident occurs on public transportation. Of the 31.8% of the respondents who believed they had the ability to deal with fires on public transportation, there was no statistically significant association between their self-reported ability to deal with fires and accuracy of their fire response knowledge.

## 4. Discussion

The metro system is reported to be the most widely used daily transportation in Hong Kong. Meanwhile, private cars and buses are respectively perceived as the safest and the least safe transportation. Around one-third of respondents were worried that a disaster/incident will occur on their daily transport, in particular for the female gender, people receiving CSSA, and unmarried people. Consistent to the findings in another study in the same community [15], the respondents’ knowledge accuracy was relatively low in the community. People with a higher educational level were more likely to report fire response knowledge that were either included in first-aid training or promoted by the local Fire Service Department. Older people and married individuals were more likely to correctly answer the fire response question which is not commonly included in first-aid handbooks or the promotion materials for fire responses. Only one-third of the respondents have received first-aid training and those with a higher educational level were more likely to have been trained. Approximately two-third of respondents were willing to learn about community disaster preparedness and respondents who expressed worries were more willing to learn.

Despite the 2017 MTR fire accident, the public’s confidence in the MTR remained higher than buses, which is the other major mode of public transportation Hong Kong. Around 85% still considered the MTR as a safe mode of public transportation while only 79% agreed buses were safe. According to the Traffic Report 2016, public buses had the highest accident rate with 394 accidents per 1000 licensed vehicles. In the same year, nine percent of the bus accidents involved other vehicles (*n* = 2261) and majorities of the impact resulted in minor injuries (*n* = 1981) rather than serious ones (*n* = 276) or fatalities (*n *= 13) [24], which is consistent with the perceived safety from the study respondents. Another possible reason for the higher perceived risk of bus accidents may be due to previous major bus crashes and their media coverage, including the 2003 Tuen Mun Road Ting Kau bus accident resulting in 21 deaths and 20 injured [25] and 2008 Sai Kung Nam Wai Road bus crash which caused 18 deaths and 44 injured [26].

In this study, private cars were rated as the safest mode of transport. Yet, in Hong Kong, private cars caused more number of road accidents than trains and buses [24]. Savage’s 2013 study about the United States also reported higher fatality risks for private cars relative to mainline trains, buses, and commercial aviation with the relative risks respectively at 17, 67 and 112 times [27]. A gap between people’s risk perception and actual risk of private car safety has been observed in this study. However, the choice of transport is complex and was found not directly associated with the perceived safety level, which is consistent with previous studies [28,29,30]. Of note, the percentage of people choosing walking/ cycling was the highest among non-public transport modes. This may be associated with the increasing awareness in environmental protection in recently years. Other factors such as worry about unpleasant incidents [29], perceived control and trust in authorities [30] as well as other economic, convenience, and comfort factors come into deciding which transportation to take.

Among the 35% of respondents who expressed worry about disaster/incidents occurring during their daily transport experiences, the female gender, those receiving CSSA and unmarried people were found to be significant predictors. Women reported more worry than men, which is consistent with research which looked at gender differences in risk perception [31,32]. The result which showed people who received CSSA were more worried about disasters was also consistent with the finding that poverty is likely to be associated with frequent accidents and mental disorders [33]. On the other hand, it was uncertain why unmarried people expressed more concern about disaster/incidents on transportation, though Dugas and Robichaud suggested that individuals who are unmarried or divorced, receiving disability payments, and have very low annual incomes are associated with Generalized Anxiety Disorder [34].

Given perceived safety level appeared to have mild impact on the choice of transport mode, learning how to response to emergencies seems to be a good way to reduce health risks and dispel worries. First-aid training is a vital building block to the enhancement of personal disaster preparedness to Health-EDRM, and first-aid training was found to be positively associated with greater perceived knowledge on how to handle a medical emergency and demonstrate first-aid skills [8]. In this study, the relationship of first-aid training and knowledge accuracy in health risk management showed a mixed pattern. Those who had previously received first-aid training were more likely to correctly answer the first-aid related fire response question (T1), whilst no association or negative association were found for the accuracy of the other questions. Despite the mixed pattern, first-aid training was shown to be having beneficial effects in building fire emergency response knowledge among the general public. However, only 32.0% of the study respondents had previously received first-aid training and thus may potentially explain the small proportion of respondents who believed they are capable of responding to fire incidents on public transport and disasters in general. Promoting knowledge of emergency response (such as first-aid, general fire response as well as electrical fire response) and increasing awareness of personal vulnerability can be crucial for disaster preparedness in urban cities [35]. Other studies conducted that targeted specific sub-groups established similar conclusions [36].

The percentage of respondents who received first-aid training in a 2017 survey (32.0%) was slightly higher than that in 2012 (26.1%) [11]. The two most important reasons of receiving first-aid training were the relevance to job duty (39.1%) and personal interest (34.9%). First-aid training was also found to be associated with a higher educational level, which is consistent with previous studies [8,37]. However, the percentage of participants who received first-aid training in Hong Kong was much lower than Norway (90%) [38], Germany (80%) and Austria (80%) [39]. In Norway, first-aid training is part of the national school curriculum for grades 7 and 10, and is required by law for some occupations, such as drivers and employees in schools and kindergartens [38]. In addition, the gap between the low percentage of respondents who received first-aid training (32%) and willingness to learn more about disaster response (67.6%) indicates that urban residents, despite of the information access and resource availability, are inadequately prepared for individual self-help skills in Health-EDRM during emergencies and disasters. This finding suggests that there might be a gap and need in emergency response training.

Study limitations include the inability of cross-sectional studies to draw causative conclusion in their design. In addition, households which were not subscribed to land-based telephone service may be missed. However, the penetration rate of residential fixed line service in Hong Kong is more than 90% [40], which implied that most households have at least one home-based telephone a number of previous studies managed to provide valuable scientific evidence to the research community with telephone survey studies [41]. The valid sample size of 1000 and the comparability of the sample population with census data (stratified by key sociodemographic characteristics) will support the potential generalizability of the research findings. Furthermore, data collected in this study was based on self-reporting, which makes it difficult to validate the accuracy of the answers. The limited amount of time in each telephone survey had also restricted our ability to examine more detailed answers and the close-ended questions may limit answers from the respondents. Nevertheless, as the field data collection was completed within a short period after the subway fire incident, there should be minimal recall bias. According to the Travel Characteristics Survey 2011 [42], metro systems accounted for 30% for all trip purposes, while bus accounted for 49%. Although these results were different from our findings, it is possibly because of the expansion of the Hong Kong subway network after 2011. For instance, from 2014 to 2016, ten additional new transfer stations became operational. Therefore, despite the design limitations, the study findings provide evidence and an overview of how the urban residents may perceive their own health-related emergency risks as well the current attitude and knowledge gaps which might affect health and safety promotion in an urban community in Asia.

## 5. Conclusions

This study provides updated scientific evidence on general urban risk perception and Health-EDRM in a public transportation system [15]. In general, subway was the most popular public transport and respondents thought it was safe despite the event of a severe MTR fire accident. Perceived confidence in handling fire on transportation and fire response knowledge were relatively low. Previous first-aid training, the proxy indicator assess individual skills in Health-EDRM, was found to be an associating factor of better first-aid related fire response knowledge. However, the proportion of respondents who had previously received first-aid training was low. More than half of the respondents showed a willingness to learn more about community disaster preparedness, especially for those who expressed worry about transport safety. The promotion from local authorities about relevant knowledge and training on first-aid and other emergency response and preparedness activities may raise awareness, increase capacity for self-help, and reduce adverse health risks in times of emergencies and crisis, especially for people with low education level.

## Figures and Tables

**Figure 1 ijerph-16-00228-f001:**
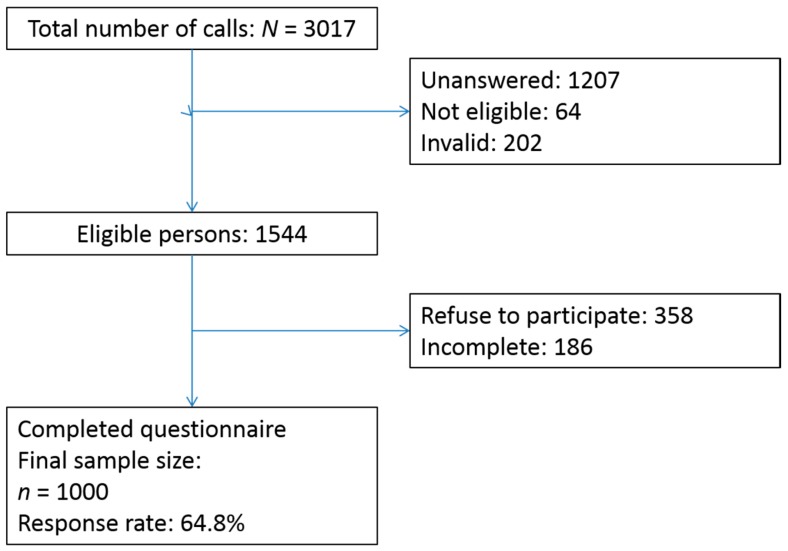
Study flow of the telephone survey.

**Figure 2 ijerph-16-00228-f002:**
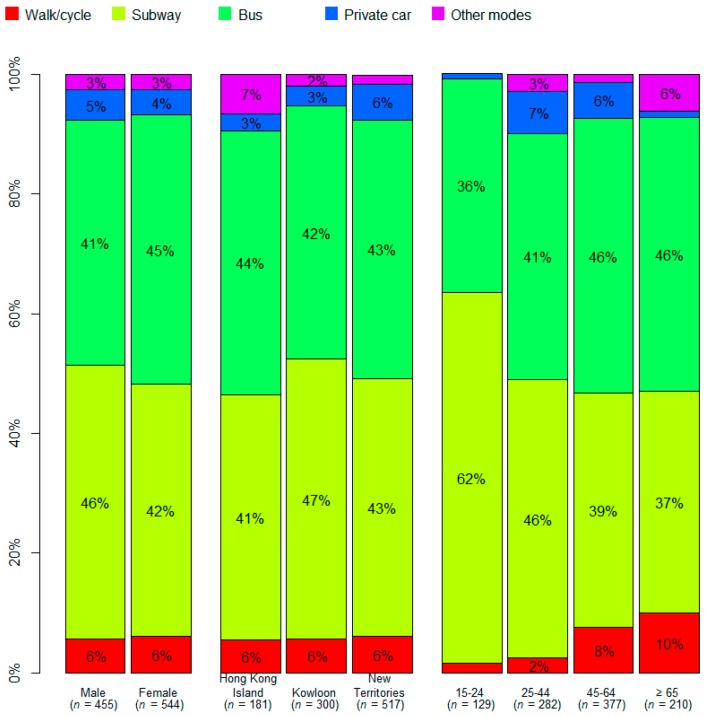
Subgroup analysis on daily transport mode.

**Table 1 ijerph-16-00228-t001:** Sociodemographic characteristics of the survey respondents in March 2017 and the general population in Hong Kong in 2016.

Demographics		Sampled Respondents (*N* = 1000)		HK 2016 Population By-Census Data (*N* = 6,506,130)		Sample vs. Census *p*-Value ^a^
		*n*	%	*n*	%	
Gender	Male	456	45.6%	2,947,073	45.3%	0.87 ^b^
	Female	544	54.4%	3,559,057	54.7%	
Age	15–24	129	12.9%	785,981	12.1%	<0.01
	25–44	283	28.3%	2,228,566	34.3%	
	45–64	377	37.7%	2,328,430	35.8%	
	≥65	210	21.0%	1,163,153	17.9%	
Area of residence *	Hong Kong Island	182	18.2%	1,120,143	17.2%	0.70
	Kowloon	300	30.0%	1,987,380	30.6%	
	New Territories	517	51.8%	3,397,499	52.2%	
Education attainment	Primary and below	116	11.6%	1,673,431	25.7%	<0.01
	Secondary	474	47.5%	2,841,510	43.7%	
	Post-secondary	408	40.9%	1,991,189	30.6%	
Marital status	Single	439	44.2%	2,708,709	41.6%	0.11 ^b^
	Married	554	55.8%	3,797,421	58.4%	

^a^*χ*^2^ test was used to measure the overall difference between this survey and the 2016 Hong Kong Population Census data. *p*-Value < 0.05 indicates significant difference. ^b^
*χ*^2^ test with continuity correction was used. * Marine population was excluded.

**Table 2 ijerph-16-00228-t002:** Pattern of daily transport and level of perceived safety.

	*n*	%	1 Strongly Disagree	2 Disagree	3 Slightly Disagree	4 Slightly Agree	5 Agree	6 Strongly Agree	Mean	SD
Total	996	100%	1.4%	2.0%	14.1%	22.6%	38.9%	21.1%	4.59	1.11
Walk/cycle	57	5.7%	1.8%	1.8%	14.0%	17.5%	38.6%	26.3%	4.68	1.17
Subway	437	43.9%	1.4%	0.7%	12.4%	20.8%	41.0%	23.8%	4.71	1.07
Bus	430	43.2%	1.4%	3.3%	16.3%	25.6%	36.7%	16.7%	4.43	1.13
Private car	46	4.6%	2.2%	2.2%	8.7%	19.6%	32.6%	34.8%	4.83	1.20
Other modes	26	2.6%	0.0%	3.8%	15.4%	19.2%	50.0%	11.5%	4.50	1.03

Note: The question is “My daily transport is safe”.

**Table 3 ijerph-16-00228-t003:** Factors associated with the expressed worry of disaster/incident occurring on my daily transport.

Characteristics		I am Worried that Disaster/Incident will Occur on the Daily Transport I Take (*n* = 990)			
		Stage 1 Model		Stage 2 Model	
		OR (95% CI)	*p*-Value	OR (95% CI)	*p*-Value
Gender	Male	1		1	
	Female	1.88 (1.44–2.46)	<0.01	1.92 (1.47–2.52)	<0.01
Age	15–24	1			
	25–44	1.33 (0.86–2.06)	0.20		
	45–64	0.90 (0.58–1.37)	0.61		
	≥65	1.24 (0.78–1.97)	0.36		
Area of residence	Hong Kong Island	1			
	Kowloon	0.92 (0.63–2.10)	0.66		
	New Territories	0.82 (0.58–1.16)	0.26		
Education	Primary or below	1			
	Secondary	0.91 (0.59–1.39)	0.66		
	Post-secondary or above	0.98 (0.64–1.50)	0.92		
Marital status	Single	1		1	
	Married	0.72 (0.55–0.93)	0.01	0.75 (0.57–0.98)	0.04
Form of daily transport	Walk/cycle	1			
	Subway	0.79 (0.45–1.39)	0.41		
	Bus	0.76 (0.43–1.34)	0.34		
	Private car	0.63 (0.28–1.43)	0.27		
	Others	1.23 (0.48–3.14)	0.67		
Accept Comprehensive Social Security Assistance	No	1		1	
	Yes	2.32 (1.23–4.38)	0.01	2.51 (1.30–4.83)	0.01

**Table 4 ijerph-16-00228-t004:** Factors associated with receiving first-aid training and willingness of learning more about community disaster preparedness.

Characteristics		Did You ever Receive First-Aid Training? *n* = 997				Willingness of Learning More about Community Disaster Preparedness *n* = 994			
		Stage 1 Model		Stage 2 Model		Stage 1 Model		Stage 2 Model	
		OR (95% CI)	*p*-Value	OR (95% CI)	*p*-Value	OR (95% CI)	*p*-Value	OR (95% CI)	*p*-Value
Gender	Male	1				1		1	
	Female	0.86 (0.66–1.12)	0.27			1.48 (1.13–1.93)	<0.01	1.51 (0.80–2.81)	0.20
Age	15–24	1		1		1		1	
	25–44	1.41 (0.90–2.18)	0.13	1.35 (0.87–2.11)	0.18	1.53 (0.98–2.40)	0.06	3.92 (0.95–16.09)	0.06
	45–64	1.03 (0.67–1.58)	0.9	1.29 (0.83–2.00)	0.26	1.14 (0.75–1.74)	0.55	1.08 (0.41–2.86)	0.88
	≥65	0.57 (0.35–0.94)	0.03	1.05 (0.61–1.79)	0.87	0.78 (0.49–1.23)	0.28	0.66 (0.23–1.91)	0.45
Area of residence	Hong Kong Island	1				1			
	Kowloon	1.04 (0.70–1.55)	0.86			0.96 (0.65–1.41)	0.83		
	New Territories	1.14 (0.79–1.64)	0.48			1.15 (0.80–1.64)	0.46		
Education	Primary or below	1		1		1		1	
	Secondary	3.96 (2.06–7.60)	<0.01	3.59 (1.83–7.07)	<0.01	1.40 (0.92–2.14)	0.12	1.45 (0.64–3.30)	0.38
	Post-secondary or above	6.82 (3.55–13.08)	<0.01	5.97 (2.96–12.03)	<0.01	1.60 (1.04–2.46)	0.03	2.37 (0.84–6.63)	0.10
Marital status	Single	1				1			
	Married	0.94 (0.72–1.23)	0.64			0.85 (0.65–1.12)	0.24		
Accept Comprehensive Social Security Assistance	No	1		1		1			
	Yes	0.42 (0.19–0.97)	0.04	0.65 (0.27–1.52)	0.32	1.08 (0.54–2.16)	0.83		
Worried about disaster/incident on the daily transport	No	1				1		1	
	Yes	1.08 (0.54–2.16)	0.3			2.93 (2.14–4.00)	<0.01	4.36 (1.69–11.25)	<0.01

**Table 5 ijerph-16-00228-t005:** Knowledge test of fire emergency response.

Fire Response Questions	Overall (*n* = 981)		Do not Know how to Deal with Fire in Transport (*n* = 627)		Know how to Deal with Fire in Transport (*n* = 354)		OR (95%CI) of Getting a Correct Answer (Know how to Deal with Fire vs. Do not Know)
	Incorrect	Correct	Incorrect	Correct	Incorrect	Correct	
T1: Room temperature water	53.0%	47.0%	52.7%	47.3%	53.3%	46.7%	OR = 0.98, 95% CI: 0.75–1.27, *p* = 0.86
T2: Fire blanket	38.7%	61.3%	37.4%	62.6%	40.3%	59.7%	OR = 0.91, 95% CI: 0.69–1.21, *p* = 0.52
T3: Fire Hose/ Extinguisher	46.2%	53.8%	48.1%	51.9%	44.0%	56.0%	OR = 1.18, 95% CI: 0.91–1.54, *p* = 0.22

Note: Specific question in knowledge test of fire emergency response: T1. Should room temperature water or ice water and ice be used to treat the burn? (a) ice water/ ice cube; (b) room temperature water; T2. If you are in a fire incident setting and you found someone was on fire, how would you use a fire blanket (a) put out the fire directly with the fire blanket; (b) cover the victim with the fire blanket and ask them to roll until the fire stops); T3. If there is no fire blanket at the scene, should fire hoses or extinguishers be used on people) (a) Yes; (b) No.

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
