# Peer review of "Health Emergency Disaster Risk Management of Public Transport Systems: A Population-Based Study after the 2017 Subway Fire in Hong Kong, China"

_ijerph, 2019, doi:10.3390/ijerph16020228_

Round 1

Reviewer 1 Report

The manuscript presents an interesting result using cross-sectional telephone survey, descriptive statistic analysis, Person’s x2 test, and logistic regression. The methods are properly used and 1000 sample is ok. I think the V1 version is good to go after a minor revison.  

1 Give the full names for EDRM in the first place in abstract.

2 Line 90-91, “a total of 12 questions were asked to 90 solicit respondents were” is not correct.

3 Line 179, “were less likely to less likely to answer”, check it.

4 In Table 5, why variables with P-value more than 0.1 go to the final model? According to your statement in line 150-151, only that less than 0.1 will.

5 Check Line 197. “…which the was consistent…”.

Author Response

Reviewer 1

The manuscript presents an interesting result using cross-sectional telephone survey, descriptive statistic analysis, Person’s x2 test, and logistic regression. The methods are properly used and 1000 sample is ok. I think the V1 version is good to go after a minor revison.

Response: Thank you for your comments and pointing out all these mistakes. We have revised the paper accordingly. Please kindly see the below.

1 Give the full names for EDRM in the first place in abstract.

Response: Thanks for your reminder. We have added this to the text.

2 Line 90-91, “a total of 12 questions were asked to 90 solicit respondents were” is not correct.

Response: Thanks for pointing this out. We have corrected the grammatical mistake.

3 Line 179, “were less likely to less likely to answer”, check it.

Response: Thanks and we have corrected this.

4 In Table 5, why variables with P-value more than 0.1 go to the final model? According to your statement in line 150-151, only that less than 0.1 will.

Response: Thanks. It is a typing mistake. We have corrected the table. We confirmed only variable with p-value < 0.1 in the Chi-square test were entered into the final model.

5 Check Line 197. “…which the was consistent…”.

Response: Thanks. We have amended this.

Reviewer 2 Report

The authors explore the relation notably between confidence in public transportation, risk perception and socio-demographic characteristics... If this subject can be an interesting one, the authors lack arguments explaining why it can. 

For example, in the introduction section they gave a lot of examples of subway catastrophies, but not really justify the dimensions they explored (confidence, risk perception, etc.). Have they some hypotheses ? What would those dimensions possibly explain ?  ...

In the "Materials and methods" section, authors should better describe the measuring scales they used. They should also give the complete list of questions asked. The reader will then be more able to understand the data. 

In the "discussion" section, participants should more comment their data. For example, how do they explain the different levels of risk perception related to the different transports mode ?  

But for the moment the main problem of this text is the low level of the writing in english (line 41, 43, ........140, 153, 164, etc.....). It is sometimes complicated to undestand what the authors wanted to express. 

In short, the text needs to be improved in form and content.

Author Response

Reviewer 2

The authors explore the relation notably between confidence in public transportation, risk perception and socio-demographic characteristics... If this subject can be an interesting one, the authors lack arguments explaining why it can. 

Response: Thank you so much for your questions, comments, and suggestions. We would take this opportunity to clarify the needs and the aim of this study.

A large number of people travel through public transport systems in cities every day. Transport safety is an important public health concern as any incidences happened within the system will be likely lead to large health impacts. In case there any emergencies within the transport system, an appropriate personal response may lower risks and even save lives. Therefore we focused on personal emergency response in this study by assessing possible associating factors related to this such as risk perception, previous trainings and other personal characteristics. We aim to identify the factors that could improve personal emergency response and we hypothesized mode of transport (which is related to risk perception), socio-demographic characteristics and first aid training may affect the awareness or knowledge of emergency response. We found that the mode of transport used and socio-demographic characteristics were less associated awareness and knowledge of emergency responses. Previous first-aid training was found associated in response knowledge. This suggests emergency training, for instance first-aid in this study, is a considerable solution to raise relevant knowledge among the general public in facing emergencis. The study also found that half of the respondents, regardless of socio-demographic characteristics, were willing to learn more about the emergency response. This shows the positive attitude of the community. Thus, findings of this study should provide insights and evidence for the emergency and disaster risk reduction management in the personal prevention level.

We have added this to the introduction to emphasize the aim and hypothesis from line 31.

“In case there are any emergencies within the metro system, appropriate personal response may lower health risks and even save lives. Therefore, this study focused on personal emergency response by assessing possible associating factors such as risk perception, previous trainings and other personal characteristics. We aimed to identify the factors that could improve personal emergency response and hypothesized that different modes of transport (which is related to risk perception), socio-demographic characteristics, and first-aid training may affect the awareness or knowledge of emergency response.”

For example, in the introduction section they gave a lot of examples of subway catastrophies, but not really justify the dimensions they explored (confidence, risk perception, etc.). Have they some hypotheses ? What would those dimensions possibly explain ?  ...

Response: The catastrophes raised in the Introduction were to give some examples of incidences that happened within subway systems that had caused large health impacts. Since they happened in different contexts and no individual details information was available, they were not for hypothesis here and were only for supporting the existence and the severity of the risk. We have pointed this out following the examples in line 15:

“Ensuring public safety, education, and emergency preparedness could have been immensely important to reduce potential harm during and immediately after an emergency incident.”

In the "Materials and methods" section, authors should better describe the measuring scales they used. They should also give the complete list of questions asked. The reader will then be more able to understand the data. 

Response: Thanks for this suggestion and we agree that this will help readers in understanding the scale better. We had included the questions asked in the appendix as suggested. Thank you.

In the "discussion" section, participants should more comment their data. For example, how do they explain the different levels of risk perception related to the different transports mode?  

Response: Thanks for your comment. We have discussed the different levels of risk perceptions between the two main public transport modes, subway and buses, in the second paragraph in the discussion using both local and international examples. We have also added a couple of sentences highlighting the relative higher safety rating of private cars in this study. Line 206.

“In this study, private cars were rated as the safest mode of transport. In fact, in Hong Kong, private cars caused more number of road accidents than trains and buses [21]. Savage’s 2013 study about the United States also reported higher fatality risks for private cars relative to mainline trains, buses, and commercial aviation with the relative risks respectively at 17, 67 and 112 times, [24]. However, the choice of transport is complex and was found not directly associated with the perceived safety level, which is consistent with previous studies [25, 26, 27].  Other factors such as worry about unpleasant incidents [26], perceived control and trust in authorities [27] as well as other economic, convenience, and comfort factors come into deciding which transportation to take. “

But for the moment the main problem of this text is the low level of the writing in english (line 41, 43, ........140, 153, 164, etc.....). It is sometimes complicated to undestand what the authors wanted to express. 

Response: Thanks for your comments. We have edited the manuscript as suggested.

In short, the text needs to be improved in form and content.

Round 2

Reviewer 2 Report

On which litterature do the authors based their hypothesis (Line 60 to 67) ? There are no references at all. We still don't know why they choose those indicators (risk perception, demographic information, etc.) and no others. They certainly have good reasons but we need to know their justification.

More over, the meaning of the results is difficult to understand. The questions that have been asked are very vague. What is the meaning of "perceived safety" in this context ? Better understand this point is of prior importance because safety may be associated with numerous dimensions (safety for environment, safety for oneself in terms of technical problems, safety for oneself in terms of agressive behavior, etc. ). At is, it is not clear what is behind this notion of safety.  
For example why is " a fire" set appart from "disaster/incident". Isn't a fire a disaster ?

Author Response

Comments and Suggestions for Authors

On which litterature do the authors based their hypothesis (Line 60 to 67) ? There are no references at all. We still don't know why they choose those indicators (risk perception, demographic information, etc.) and no others. They certainly have good reasons but we need to know their justification.

Response to comment:

Thanks for your comment. We have included the rationale for the hypothesis with supporting literature in the Introduction.

More over, the meaning of the results is difficult to understand. The questions that have been asked are very vague. What is the meaning of "perceived safety" in this context ? Better understand this point is of prior importance because safety may be associated with numerous dimensions (safety for environment, safety for oneself in terms of technical problems, safety for oneself in terms of agressive behavior, etc. ). At is, it is not clear what is behind this notion of safety.  
For example why is " a fire" set appart from "disaster/incident". Isn't a fire a disaster ? 

Response to comment:

Thanks for your suggestions. We have defined “perceived safety” as suggested to make the results less confusing.

“Perceived safety” here meant respondents’ rating on the safety level of the transport mode they used daily.

Fire is one of the most common hazards in transport disasters. We, therefore, selected fire-response as one of the focuses in this study. We have included this in the Introduction part.

We thanks and appreciate your useful comments and suggestions for improving the quality of this paper.